# Disadvantaged Americans are suffering the brunt of rising pain and physical limitations

**Dana A. Glei** *, **Maxine Weinstein**

Center for Population and Health, Georgetown University, Washington, DC, United States of America

* dag77@georgetown.edu

**Data Availability Statement:** All of the data used in this analysis are publicly available. The NHIS can be downloaded from https://www.cdc.gov/nchs/nhis/index.htm. MIDUS is available from ICPSR at https://www.icpsr.umich.edu/web/ICPSR/series/

## Abstract

Using data from three national surveys of US adults (one cohort and two cross-sectional studies, covering the period from the mid-1990s to the mid-2010s), we quantify the degree to which disparities by socioeconomic status (SES) in self-reported pain and physical limitations widened and explore whether they widened more in midlife than in later life. Unlike most prior studies that use proxy measures of SES (e.g., education), we use a multidimensional measure of SES that enables us to evaluate changes over time in each outcome for fixed percentiles of the population, thereby avoiding the problem of lagged selection bias. Results across multiple datasets demonstrate that socioeconomic disparities in pain and physical limitations consistently widened since the late 1990s, and if anything, widened even more in midlife than in late life (above 75). For those aged 50–74, the SES disparities in most outcomes widened by more than 50% and in some cases, the SES gap more than doubled. In contrast, the magnitude of SES widening was much smaller above age 75 and, in the vast majority of cases, not significant. Pain prevalence increased at all levels of SES, but disadvantaged Americans suffered the largest increases. Physical function deteriorated for those with low SES, but there was little change and perhaps improvement among the most advantaged Americans. At the 10th percentile of SES, the predicted percentage with a physical limitation at age 50 increased by 6-10 points between the late-1990s and the 2010s, whereas at the 90th percentile of SES, there was no change in two surveys and in the third survey, the corresponding percentage declined from 31% in 1996–99 to 22% in 2016–18. The worst-off Americans are being left behind in a sea of pain and physical infirmity, which may have dire consequences for their quality of life and for society as a whole (e.g., lost productivity, public costs).

## Introduction

Since the early 1990s, the US has exhibited widening socioeconomic disparities in the prevalence of pain [1–5] and physical function [4,6,7]. Rising pain and growing physical limitations may cause diminished quality of life, higher health care costs, reduced earning capacity, increased reliance on disability benefits, and greater need for long-term care. Since 1985, the number of disabled-worker beneficiaries has more than tripled [8]. Among disabled Medicare

203/; in this analysis, we use the data from the 1995-96 MIDUS 1 (https://doi.org/10.3886/ICPSR02760.v19) and 2011-14 MIDUS Refresher (https://doi.org/10.3886/ICPSR36532.v3) waves. Data from the HRS are available from https://hrs.isr.umich.edu/data-products.

**Funding:** This work was supported by the National Institute on Aging [https://www.nia.nih.gov/, grant numbers P01 AG020166 and U19AG051426 to Carol Ryff (PI of the MIDUS project)] and the Graduate School of Arts and Sciences, Georgetown University (https://grad.georgetown.edu/). The funders had no role in study design, data collection and analysis, decision to publish, or preparation of the manuscript.

**Competing interests:** The authors have declared that no competing interests exist.

beneficiaries younger than 65 in 2007–11, nearly two-thirds were diagnosed with a musculo-skeletal condition such as back, neck, or joint pain [9]. Disadvantaged Americans have been disproportionately affected by rising pain and deteriorating physical function [1–7].

Some evidence suggests that the trends in pain and physical function may also vary by age. Prior studies indicate that the recent rise in physical limitations/disability in the US [10] may be limited to midlife Americans—particularly below age 65 or 70 [11–13]. Similarly, Zimmer and Zajacova [14] showed that the increases in mild/moderate and severe/limiting pain were greater for those aged 55–71 than for those aged 72 and older. The findings, however, are not fully consistent: Zajacova et al. [5] found that the relative increase in pain prevalence over time was somewhat steeper at ages 45–84 than at ages 25–44.

To our knowledge, no study has explicitly evaluated whether widening of SES disparities in pain and physical limitation varies significantly by age. Nonetheless, a comparison of results for pain across studies suggests that disparities may have widened more at younger ages than at older ages. Among Americans aged 25–74 [4], the SES disparity in pain widened. In con-trast, among an older sample (55+) (10), increases in severe or limiting pain were bigger for those with *more* education and wealth, implying that the SES disparity narrowed among older Americans. Other results [2] also suggest that the educational gap in knee pain and back/neck/joint pain may be larger at ages 45–64 than above age 70.

Most prior studies of SES disparities in pain and physical function [1,2,5–7,14] have used proxy measures that capture only one dimension of SES and thus, cannot represent the overall effect of SES, which is, by definition, a multidimensional construct. Some studies [1,2,7] have relied solely on education, which is prone to the problem of lagged selection bias [15]: as edu-cation levels have increased over time, high school dropouts have become a rare and select seg-ment of the US population. Other researchers entered various SES indicators (e.g., education, income, wealth) one at a time [5,6,14], but that strategy does not capture the overall effect of SES. Bravemen et al. [16] convincingly argues that education is not a good substitute for income (or vice versa), nor is income an adequate proxy for wealth. A composite measure of SES is more likely to capture the complexity of inter-related factors that determine one's access to collectively desired resources [17,18].

In this paper, we replicate analyses of SES widening in self-reported pain and physical limi-tations across three nationally-representative samples of US adults that cover both younger and older Americans. Replication across multiple datasets helps establish that the pattern is robust and not merely a fluke that reflects sample selectivity, measurement error, or random noise. First, we examine the consistency and magnitude of SES widening across surveys. To the extent possible, we harmonize the outcome measures across datasets. We use a multidi-mensional measure of SES to evaluate changes over time in each outcome for fixed percentiles of the population, thereby avoiding the problem of lagged selection bias [15]. [In this paper, we use the term "change" to refer to a period difference in aggregate-level values rather than within-individual changes.] Second, we investigate the nature of the underlying trends that generated widening SES disparities. Did the disparity widen because health deteriorated more for those with low SES, because health improved more for those with high SES, or because pain/limitations increased among those with low SES while decreasing for those with high SES? Finally, we stratify the samples into three age groups that can be compared across surveys (i.e., 25–49, 50–74, and 75+) and explore whether the magnitude of SES widening is greater in midlife than in later life. Given that mortality decline slowed or reversed among working-age (25–64) Americans [19] whereas those above 65 continued to exhibit gains in life expectancy [20], we suspect that increased pain and physical limitations may also be concentrated in mid-life. Thus, we hypothesize that the gap in SES disparities widened more at younger ages than among older Americans.

Our results demonstrate that socioeconomic disparities in pain and physical limitations consistently widened since the late 1990s. Pain prevalence increased at all levels of SES, but disadvantaged Americans suffered much bigger increases. Physical function deteriorated for those with low SES, but there was little change and perhaps improvement among the most advantaged Americans. The findings also suggest that disparities in physical limitations and overall pain widened more in midlife than in late life.

## Methods

### Data

We used data from multiple surveys with overlapping age ranges: the National Health Interview Survey (NHIS) covers the full adult age range (18+), the Midlife in the US (MIDUS) study covers midlife (25–74), and the Health and Retirement Survey (HRS) covers older Americans (50+). [The MIDUS targeted persons aged 25–74, but a few of the eventual respondents were outside that age range (see footnotes to Table A in S1 Appendix for details).] Table A summarizes the sample designs, response rates, and analysis sample for each dataset. We used the 1997–2018 Waves from NHIS ($N$ = 668,526 aged 18 and older), a cross-sectional study based on a geographically-clustered, probability sample that is representative of the civilian, non-institutionalized US population [21]. From MIDUS, we used data from the cross-sectional waves fielded in 1995–96 [22] and 2011–14 [23], each of which targeted a national probability sample of non-institutionalized, English-speaking adults aged 25–74 in the contiguous US. We restricted our MIDUS analysis to respondents who completed both the initial phone interview and the self-administered questionnaire ($N$ = 3034 in 1995–96; $N$ = 2598 in 2011–14). For the HRS cohort study, we included the 1996–2016 Waves ($N$ = 190,546 aged 50 and older), which sampled household residents of the contiguous US with longitudinal follow-up every two years [24]. HRS comprised 319,460 person-years of follow-up (mean 8.7, maximum 21 years).

### Measures

**Physical function.** We included four physical tasks that were asked in all three surveys: whether the respondent reports any difficulty lifting/carrying, climbing stairs, walking a short distance, and stooping/bending/kneeling (see Table B in S1 Appendix for details). In addition to the dichotomous variable for each individual task, we created another variable indicating whether the respondent reported any of these four limitations.

**Pain.** The pain measures varied across the three surveys (see Table C in S1 Appendix for details). The questions about specific types of body pain (e.g., headaches, back pain) were dichotomous in NHIS and HRS, but ordinal in MIDUS. We recoded the variables from MIDUS into binary indicators representing pain at least once a week.

**Relative SES.** We created the SES index based on education, occupation, income, and assets (see Text A in S1 Appendix for details). Then, we converted the index score to a percentile rank representing the individual's position within the distribution at a given wave within each survey. For ease of interpretation, we rescaled the SES variable to range from 0 (bottom percentile) to 1 (top percentile), such that a one-unit change denotes the difference between the top and bottom percentile of SES.

Unlike HRS and MIDUS, NHIS does not have a measure of wealth nor do they collect complete information about the education and occupation of the spouse/partner. Therefore, to test the robustness of the results, we re-estimated the analyses for HRS and MIDUS using only the three SES items that are comparable with NHIS (i.e., respondent's education, respondent's occupation, and income).

**Control variables.** Control variables comprise age, sex, race/ethnicity, and period (i.e., survey wave). We coded race based on the respondent's report of the group that best describes his/her racial background. Latinx ethnicity is based on self-report (e.g., "Are you of Spanish, Hispanic or Latino descent, that is, Mexican, Mexican American, Chicano, Puerto Rican, Cuban or some other Spanish origin?") with one exception: the 1995–96 wave of MIDUS did not include that question. For MIDUS, we coded Latinx ethnicity based on responses to the following question: "Other than being American, what are your main ethnic origins? That is, what countries or continents are your ancestors from?" Respondents were classified as Latinx if they reported any of the following countries/regions: Mexico, Central America, Cuba, Dominican Republic, Puerto Rico, South America (including Brazil), or Spain. Table E in S1 Appendix shows descriptive statistics for all analysis variables by survey.

## Modeling strategy

All analyses were conducted in Stata 16.1 [25]. The covariates with the most missing data were income and wealth. HRS and NHIS already imputed missing data for those variables (Text A in S1 Appendix), and there was very little missing data for other analysis variables Text B in S1 Appendix). Therefore, we did not do further imputation for those two surveys. For MIDUS, we followed standard practices of multiple imputation (see Text B in S1 Appendix for details).

Each of the binary outcomes was modeled using logistic regression model. Given evidence of a non-linear age pattern for many of the outcomes (e.g., lifting limitation, headache, joint pain), we used a quadratic specification for age. For HRS and NHIS, we compared alternative specifications for period (i.e., linear, quadratic, categorical). In NHIS, the categorical specification (i.e., 1997–99, 2000–03, 2004–09, 2010–15, 2016–18) fit best (i.e., lowest BIC) for all outcomes, whereas the results for HRS were mixed: the linear specification fit best for all the pain outcomes and most of the physical limitations outcomes, but the quadratic specification fit best for any physical limitation and stair climbing. To allow for possible non-linear period effects, we used the categorical specification in the final models. Finally, we compared linear and quintile specifications for SES. The linear specification fit best for all outcomes in HRS and MIDUS and for 8 of the 10 outcomes in NHIS. In NHIS, the quintile specification fit somewhat better for facial/jaw pain and joint pain. For the final models, we used the linear specification for SES, but we also re-estimated the models using the quintile specification as test of robustness.

All models controlled for sex, age (quadratic specification), race/ethnicity, relative SES, and period (categorical specification). We included age interactions: 1) with SES because prior research has demonstrated that disparities in health tend to narrow at older ages; and 2) with period to test whether rising pain and growing physical limitations are concentrated in midlife. We also included an interaction between SES and period to evaluate whether the period effects differed by SES.

Because we have up to 11 observations per respondent in HRS, we fit a random intercept model to account for intra-individual correlation and used the Huber-White estimator to correct the standard errors for clustering at the primary sampling unit (PSU) level. For NHIS, we used the "svy" command in Stata 16.1 to account for the complex, multi-stage sampling design (i.e., stratification, clustering, and oversampling of specific population groups).

The regression models for MIDUS and NHIS were weighted using the survey-provided weights; regression results for HRS were unweighted[When I try to use the survey weights for the random intercept model, it crashes: 1) the "xtlogit" command will NOT allow weights that vary over time for a given R; 2) the "melogit" command seems to allow it, but gives me an error message "initial values not feasible"; and 3) when I tried using the user-written gllamm

command (which is what Zimmer & Zajacova say they used), it ran for 50 minutes without converging and most of the iterations were "not concave." I ended up aborting. We have up to 11 observations per R in HRS; it is important to account for intra-individual correlation. The best way to do that for a non-linear model (such as logit) is to include a random intercept. Thus, given the choice between correcting for intra-individual clustering and using weighted regression, I would opt for the former. We *could* use the Huber-White estimator to correct the standard errors for individual-level clustering but that is less than ideal for several reasons: a) it produces different results for a non-linear model (i.e., represents the population average effect rather than the subject specific effect*); b) we are unable to correct for PSU-level clustering (as I am doing in the random intercept model). (*As Germán explains it (https://data.princeton.edu/pop510/hospmle), the population average effect is obtained by "integrating out" the random effect. In contrast, the effects from the random effect model represent a subject specific effect (i.e., conditional effect given a fixed value of the random effect). See also: https://www.stata.com/support/faqs/statistics/random-effects-versus-population-averaged/. In short, I don't think the random effects models is the *right* way to do it when you are estimating a non-linear model. For a linear model, it turns out that the population-average effect and the subject specific effect are the same.)]. Unfortunately, Stata 16.1 would not allow use of survey weights that vary over time for the same individual in a multilevel logit model with a random individual-level intercept. As a test of sensitivity, we refitted the models for MIDUS and NHIS using unweighted regression. The estimates of SES widening based on NHIS remained similar. While the estimates for MIDUS were somewhat smaller based on unweighted regression, the pattern of the results remained unchanged: the SES disparity widened significantly because pain/limitations increased more for those with low SES than for those with high SES.

Based on the model coefficients, we graphed the estimated odds ratio for the period change in SES disparity, where the SES disparity was defined as the difference between someone in the 10th versus the 90th percentile of SES. To determine the nature of the underlying trends that generated SES widening, we also plotted the odds ratios for the period change in each outcome for someone in the 10th and 90th percentile of SES. Because the model included an interaction between age and period, the magnitude of the period effect varied by age. We showed the estimates for age 50, an age that was observed in all three surveys. In Supplementary Maternal, we also plotted the estimated odds ratios for age 74, which was the oldest age observed in all three surveys.

Finally, we stratified the sample from each survey into age groups (i.e., 25–49, 50–74, and 75+) and refitted the model separately for each age group to explore whether the magnitude of SES widening was greater in midlife than in later life. To formally test whether SES widening varied significantly by age, we added a three-way interaction among age, SES, and period to the models that included the full age range for each survey. Age was treated as linear for the purposes of this three-way interaction because preliminary models indicated that specifying age as quadratic for that interaction did not improve model fit in the vast majority of cases (i.e., it did not improve fit for any outcome in MIDUS and it improved fit for only 1 of 10 outcomes in NHIS and 1 of 9 outcomes in HRS).

To visualize how these outcomes changed over time and by SES, we plotted predicted probabilities for any physical limitation and back pain at age 50 and at age 74 for a person at the 10th and 90th percentiles of SES at the beginning and the end of the time series for each survey.

## Results

All three nationally-representative surveys demonstrated widening SES disparities in physical limitations (Fig 1) and most pain measures (Fig 2). For example, the SES disparity in the

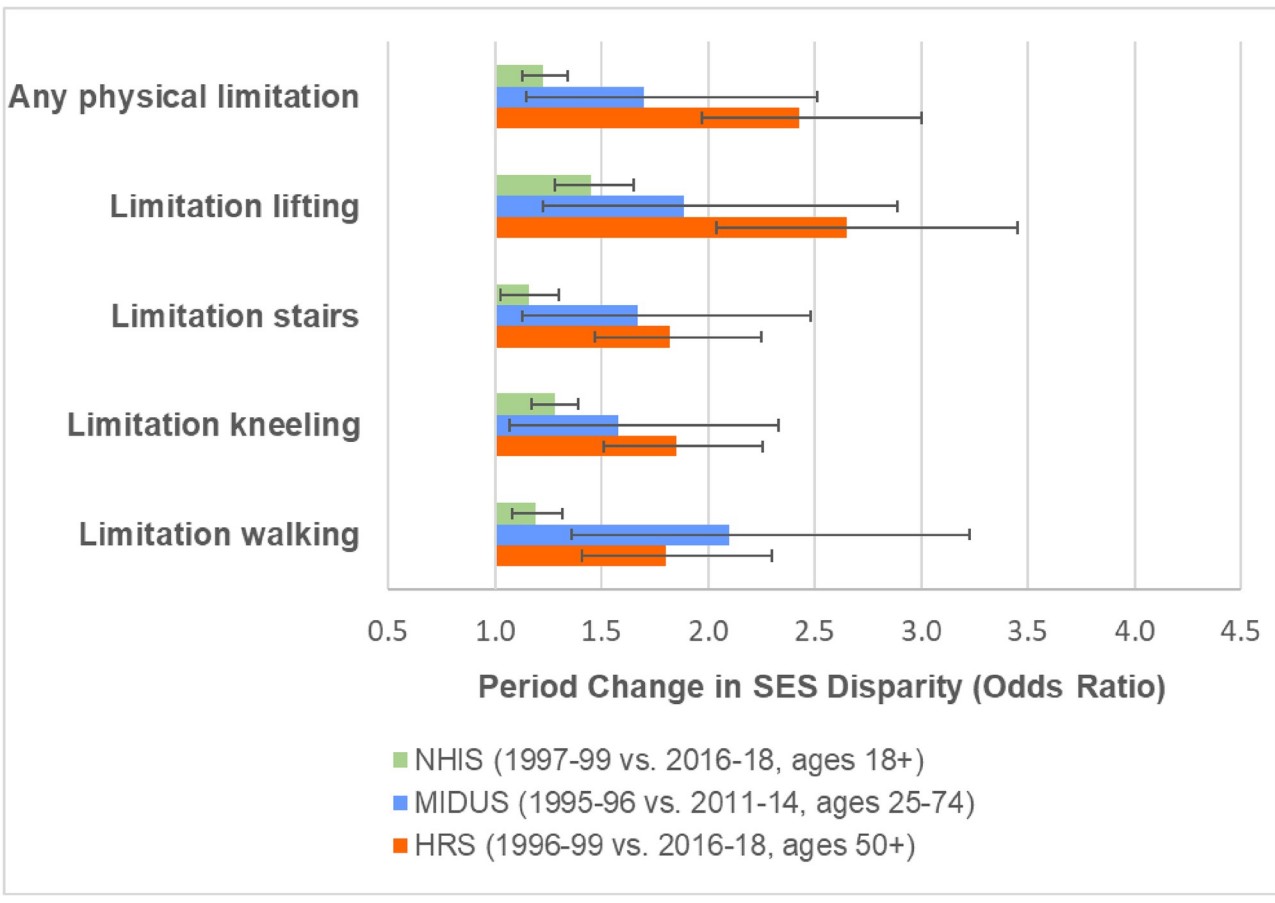

**Fig 1. Change over time in socioeconomic disparities in physical limitations.** The SES disparity was computed as the odds ratio for the 90th versus the 10th percentile of SES based on the estimates from models shown in Tables F, H, and J in S1 Appendix. All models adjusted for sex, age (quadratic specification), race/ethnicity, period (categorical specification), and SES. The models also include dthe following 2-way interactions: age (quadratic) x period; age (quadratic) x SES; and period x SES.

prevalence of any physical limitation more than doubled in HRS between 1996–99 and 2016-18. The extent of SES widening in physical limitations was generally weaker in MIDUS and weakest in NHIS. With the exception of headaches in HRS, the SES disparities in the pain outcomes also widened in all surveys for which the measure was available. Tables F-K in S1 Appendix show the full results from the models.

Figs 3–5 show the underlying period effects for the 10th versus the 90th percentile of SES that generated the widening disparities. These figures represent the values for a person aged 50; the corresponding period effects for someone aged 74 are shown in Figs A-C in S1 Appendix. Physical limitations appear to have followed a different pattern from pain. Physical limitations increased for those with low SES, whereas there was little change or perhaps even improvement for their more advantaged counterparts (Fig 3 & Fig A in S1 Appendix). In contrast, overall pain (in HRS) increased over time at all levels of SES, but the increase was larger for those with low SES (Fig 4 & Fig B in S1 Appendix). Backaches and joint pain also increased for those with low SES (Fig 5 & Fig C in S1 Appendix), but the results were mixed for those with high SES: for HRS, the increase in back pain was not significant at age 50 or 74; in MIDUS, the increase in back and joint pain was significant only at age 50; NHIS showed a

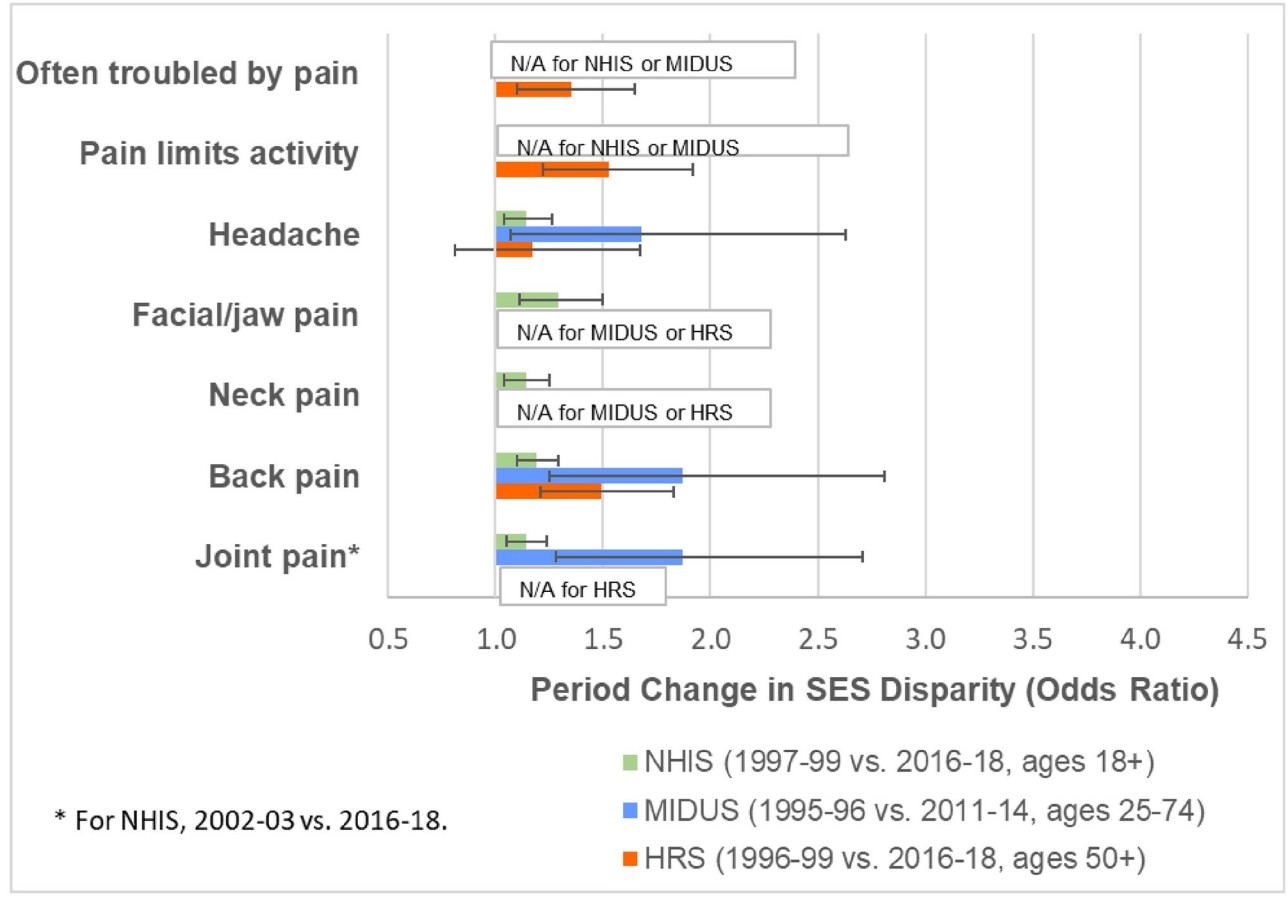

**Fig 2. Changes over time in socioeconomic disparities in pain.** Joint pain was available in NHIS only for 2002–2018. The SES disparity was computed as the odds ratio for the 90th versus the 10th percentile of SES based on the estimates from models shown in Tables G, I, and K in S1 Appendix. All models adjusted for sex, age (quadratic specification), race/ethnicity, period (categorical specification), and SES. The models also included the following 2-way interactions: age (quadratic) x period; age (quadratic) x SES; and period x SES.

significant increase in back and joint pain at both ages, although the rise was, if anything, greater at age 74 than at 50.

## Did the patterns of SES widening differ by age?

When the models were fit separately by age group, we found that the SES disparities appear to have widened more for those aged 25–74 than for those aged 75 and older. In HRS, the SES disparities in physical limitations (Fig 6), overall pain (Fig 7), and back pain (Fig 8) widened among those aged 50–74 by more than 50%: the odds ratio (OR) ranged from 1.64 (95% CI = 1.30–2.07) for back pain to 3.03 (95% CI = 2.18–4.21) for lifting. In contrast, the corresponding changes were much smaller and mostly not significant for those aged 75 and older. Although the magnitude of SES widening was smaller in NHIS, we found a similar pattern: among those aged 25–74, SES disparities grew wider for any physical limitation, limitations with lifting and kneeling, and back pain, but there were no significant changes in the SES gap for the same outcomes among those aged 75 and older. In MIDUS, the SES disparity for several outcomes (i.e., any physical limitation, kneeling limitation, headaches, back pain, and joint pain) increased significantly among those aged 25–49, while there was no significant

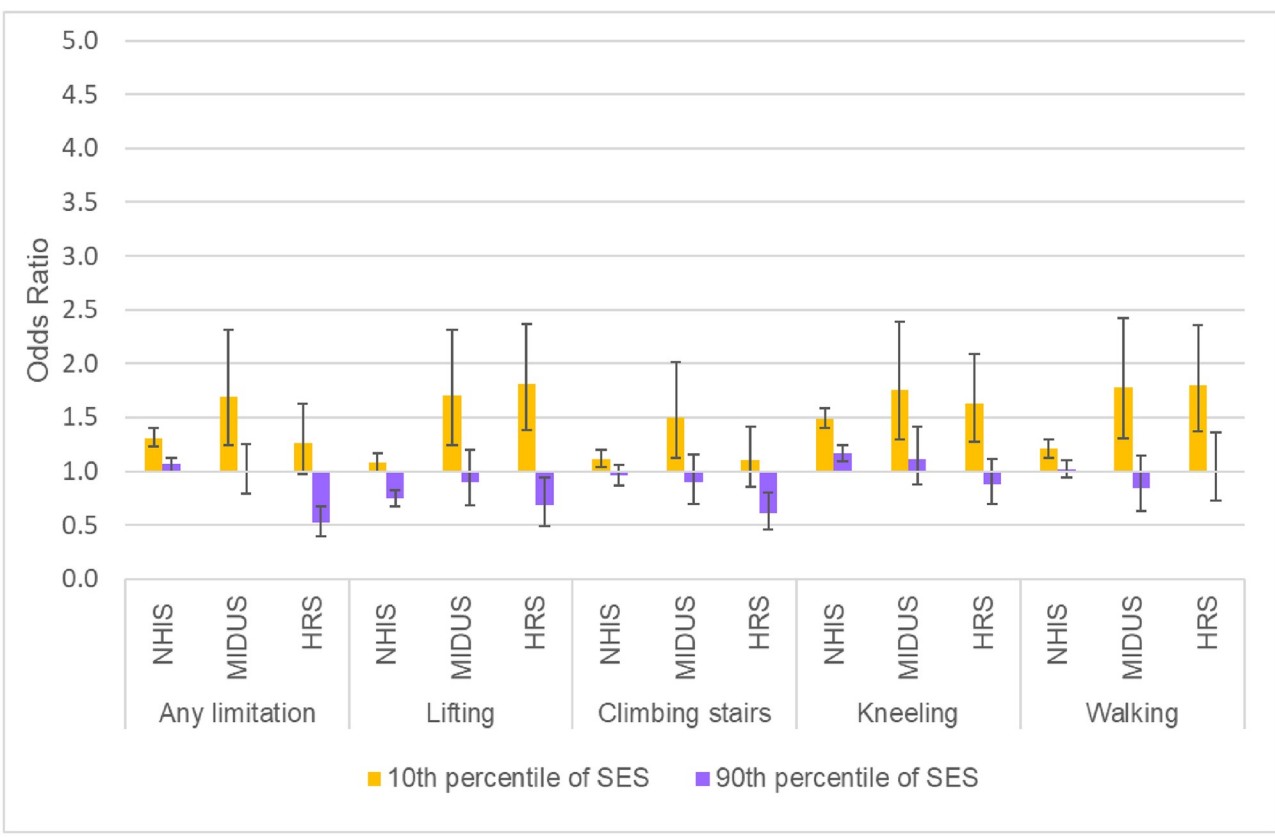

**Fig 3. Period change in physical limitations at age 50 for 10th vs. 90th percentile of SES.** These results are based on the models shown in Tables F, H, and J in S1 Appendix.

change in the SES gap among those aged 50–74 (Figs 6 & 8). Yet, for other outcomes (i.e., lifting, climbing stairs, walking) in MIDUS, the SES disparities widened as much, if not more, at ages 50–74 than at ages 25–49.

To formally test whether SES widening varied significantly by age, we added a three-way interaction among age, period, and SES to the models based on the full age range. The three-way interaction was significant for all measures of physical limitations in HRS and for kneeling limitations in both MIDUS and NHIS. The three-way interaction was also significant for the two general pain measures in HRS (these variables were not available from MIDUS or NHIS), but not for the measures of headaches, back pain, or joint pain in any of the surveys. In cases where the effect differed significantly by age, the results indicated less SES widening at the oldest ages. Thus, the analyses stratified by age group as well as the formal tests for whether SES widening varied by age suggested that SES disparities in physical limitations and overall pain widened more at younger ages than at older ages.

## Period changes in the predicted prevalence of any physical limitation for low vs. high SES

Fig 9 presents the predicted prevalence of any physical limitation for a person aged 50 with low (10th percentile) versus high (90th percentile) SES in two periods approximately two decades apart. In all three surveys, the predicted percentage with a physical limitation at age 50

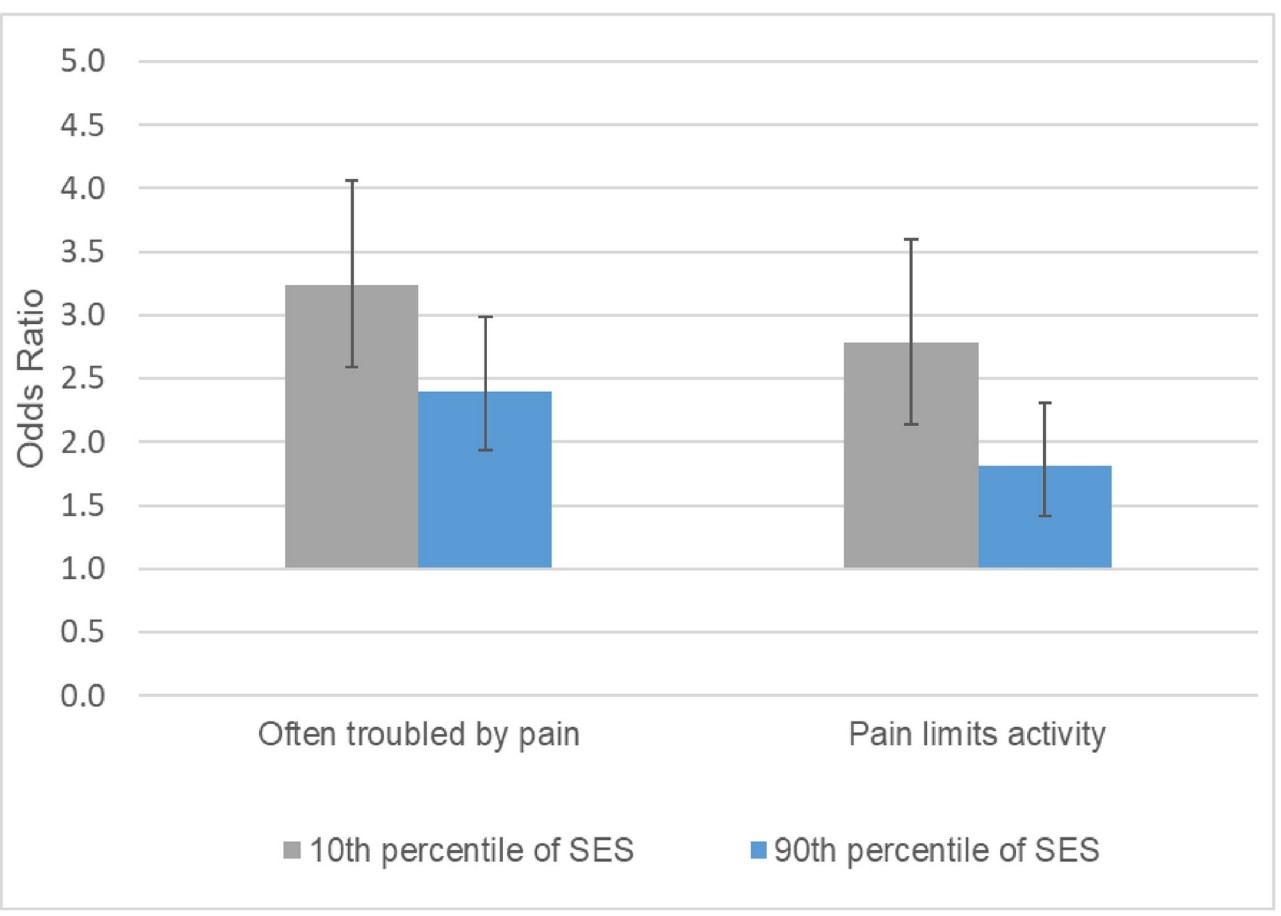

**Fig 4. Period change in overall pain at age 50 for 10th vs. 90th percentile of SES, HRS 1996–2016.** These results are based on the models shown in Table G in S1 Appendix.

increased substantially for someone with low SES: from 40% in 1997–99 to 46% in 2016–18 based on NHIS; from 68% in 1995–96 to 78% in 2011-14 based on MIDUS; and from 58% in 1996–99 to 64% in 2016–18 based on HRS). In contrast, for someone with high SES, there was virtually no change in the prevalence of a physical limitation at age 50 in NHIS or MIDUS, and in the case of HRS, the estimates indicate a decline in physical limitation (from 31% in 1996–99 to 22% in 2016–18).

The corresponding estimates for someone aged 74 are shown in Fig D in S1 Appendix. At this older age, the predicted prevalence of any limitation increased over time at all levels of SES across all surveys, but the rise in prevalence was greater at lower levels of SES than at higher SES.

## Period changes in the predicted prevalence of back pain for low vs. high SES

As shown in Fig 10, the predicted percentage with back pain at age 50 increased for those with low SES (e.g., from 31% in 1995–96 to 53% in 2011–14 based on MIDUS), but there was much less increase for those with high SES (e.g., from 17% to 21%, respectively, in MIDUS). The absolute increases in back pain were smaller in HRS and NHIS than in MIDUS. As shown in Fig E in S1 Appendix, the patterns were similar for someone aged 74.

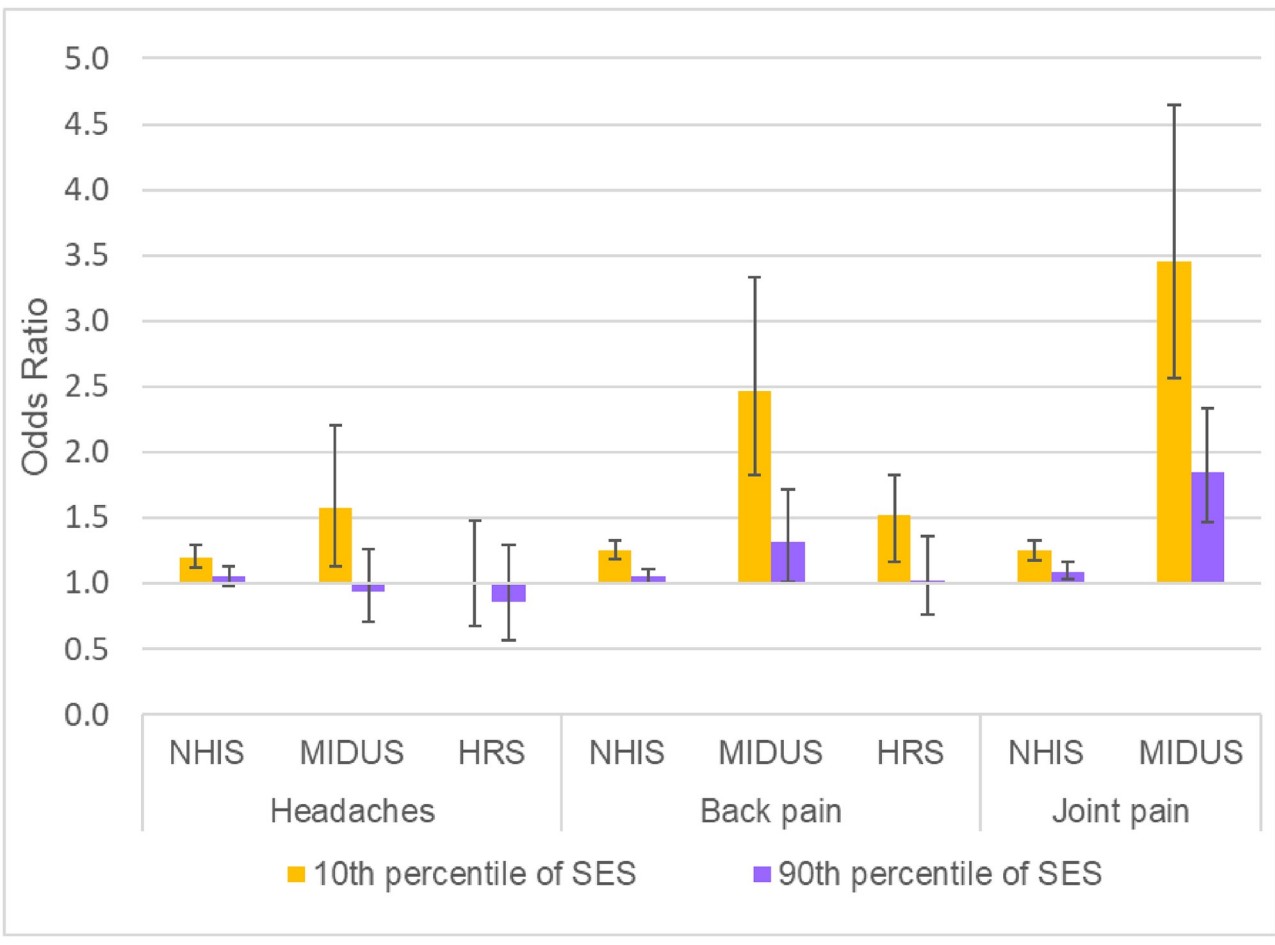

**Fig 5. Period change in specific types of pain at age 50 for 10th vs. 90th percentile of SES.** These results are based on the models shown in Tables G, I, and K in S1 Appendix.

### Sensitivity tests

When we re-estimated the models shown in Tables F-K in S1 Appendix using a categorical specification (i.e., quintiles) for SES, the results remained similar. Across all three datasets, SES disparities in physical limitations and various measures of pain widened between the mid- to late-1990s and 2010s.

We also re-estimated the models for HRS and MIDUS using the three-item version of the SES index to maximize the comparability with the SES measures in NHIS. Again, the pattern of the results remained similar although the magnitude of SES widening was often smaller. For example, the odds ratio for SES widening of any physical limitation in HRS was 2.18 (95% CI = 1.78–2.68) for the 3-item version of the SES measure versus 2.43 (95% CI = 1.97–3.00) for the full 6-item version.

### Discussion

SES disparities in pain and physical limitations consistently widened as measured across multiple datasets and a wide range of measures, and if anything, widened even more in midlife than in late life (above age 75). No prior study has demonstrated that SES widening in pain and physical limitations differs by age.

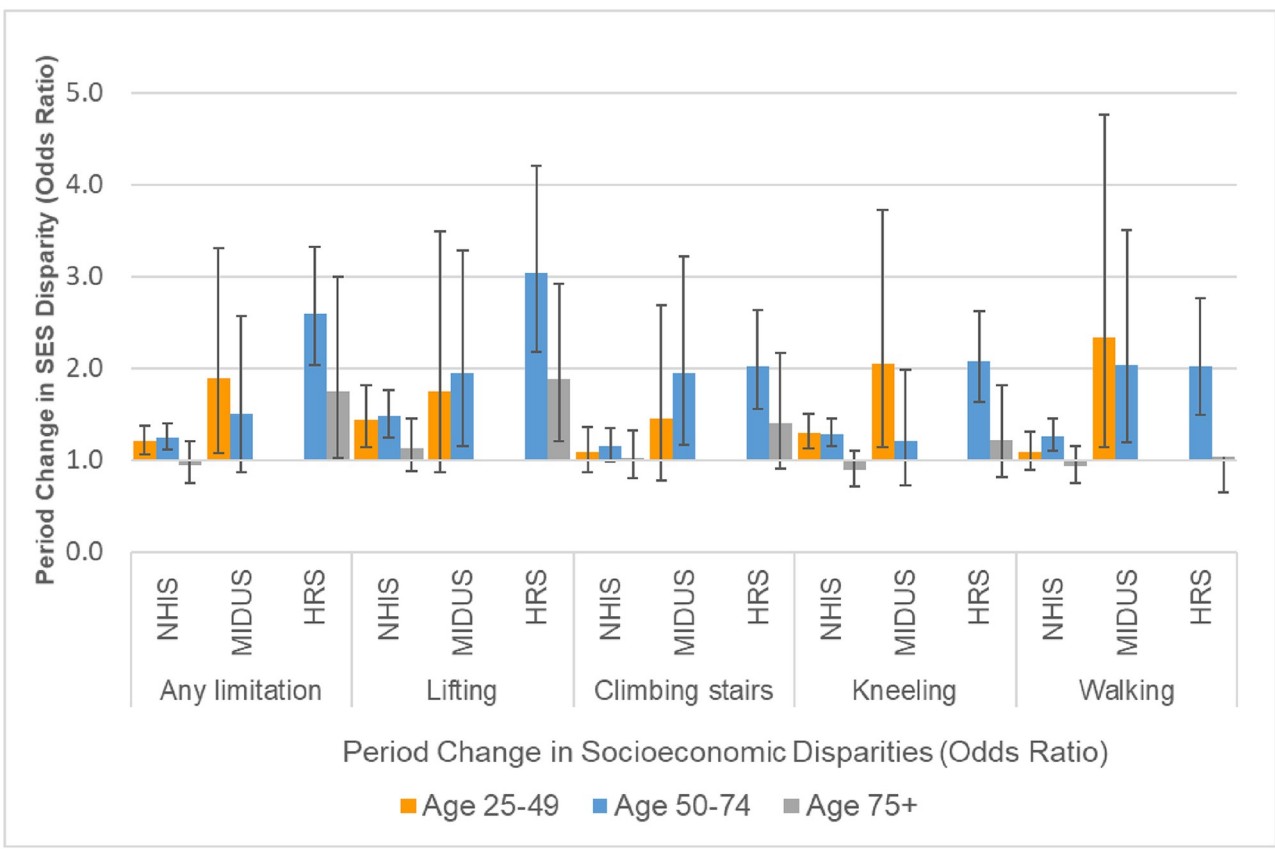

**Fig 6. Changes over time in socioeconomic disparities in physical limitations by age group.** The SES disparity was computed as the odds ratio for the 90th versus the 10th percentile of SES. The estimates were based on models fit separately for the three age groups. All models adjusted for sex, age (linear specification), race/ethnicity, period (categorical specification), and SES. The models also included the following 2-way interactions: age x period; age x SES; and period x SES.

We found that pain prevalence increased at all levels of SES, but disadvantaged Americans suffered much bigger increases. These results are consistent with previous research showing bigger increases in pain for those with lower levels of education, income, or overall SES [1–3,5]. Even advantaged Americans did not appear to be immune to rising levels of pain.

Given that numerous studies have demonstrated the strong association between pain and physical function [26–33], we expected that an increase in pain over time at the population-level would be accompanied by a corresponding decline in physical function. The results were consistent with our expectations among those with low SES (i.e., both pain and physical limitations increased over time), but we were surprised by the contradictory finding among those with high SES (i.e., overall pain increased but there was little change, or perhaps even improvement, in physical limitations).

Whereas physical function deteriorated for those with low SES, there was little change, and perhaps even improvement, among the most advantaged Americans. Prior studies [4,7] have documented increases in physical limitations among those with low education or SES, but the findings are mixed regarding the trends for those with high SES. Based on NHIS respondents aged 45–64 in 2000–2015, Zajacova & Montez [7] found slight increases in functional limitations even among those with a college degree. In the current study, we also found small increases in the prevalence of any limitation among those in 90th percentile of SES at age 74 in

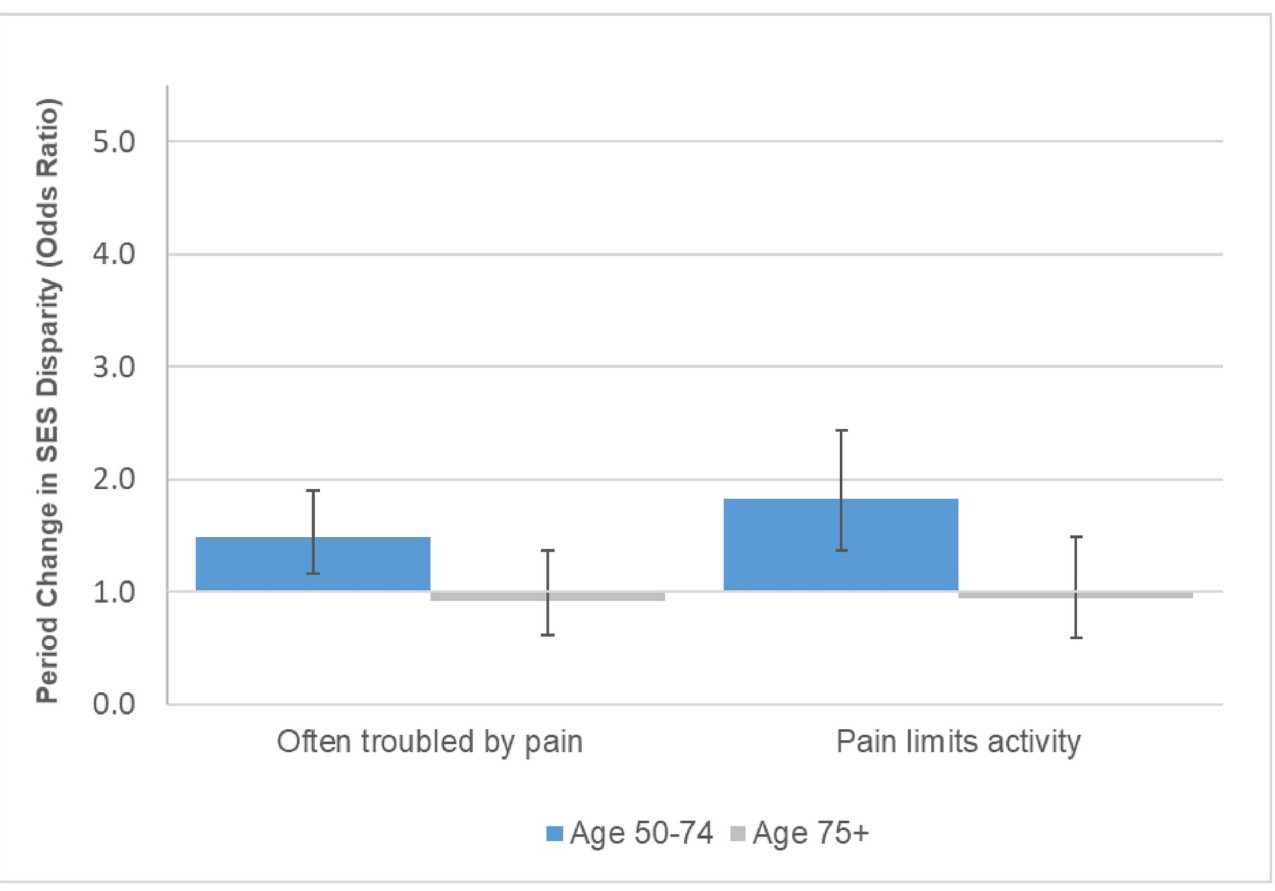

**Fig 7. Changes over time in socioeconomic disparities in overall pain by age group, HRS 1996–2016.** The SES disparity was computed as the odds ratio for the 90th versus the 10th percentile of SES. The estimates were based on models fit separately for the two age groups. All models adjusted for sex, age (linear specification), race/ethnicity, period (categorical specification), and SES. The models also included the following 2-way interactions: age x period; age x SES; and period x SES.

all three surveys. Yet, at age 50, we found little change at high levels of SES in MIDUS and NHIS, but the results from HRS indicated a significant *decline* in the prevalence of any limitation for those with high SES. Thus, we cannot say whether physical limitations improved among midlife Americans with high SES. We can only conclude that those with low levels of SES consistently exhibited more deterioration in physical function since the late-1990s than their more advantaged counterparts.

One central question, which this study cannot answer, is: <u>why</u> did pain and physical limitations increase, particularly among disadvantaged, midlife Americans? One proposed explanation focuses on the obesity epidemic [4,14,34,35]. However, the evidence regarding socioeconomic differentials in obesity trends has been mixed. An analysis based on NHIS suggested that the income disparity in obesity narrowed between 1988–94 and 2005–08, but the educational disparity widened [36]. A study based on the National Health and Nutrition Examination Survey (NHANES) reported similar increases between 1960 and 2008 in obesity across all education and income groups [37]; more recent work based on NHANES showed widening of the educational disparity in obesity over the period from 1971 to 2012, but it appeared to be driven almost entirely by white females [38]; other analyses based on NHANES showed little difference by income in the growth of obesity between 1988–94 and 2013–16

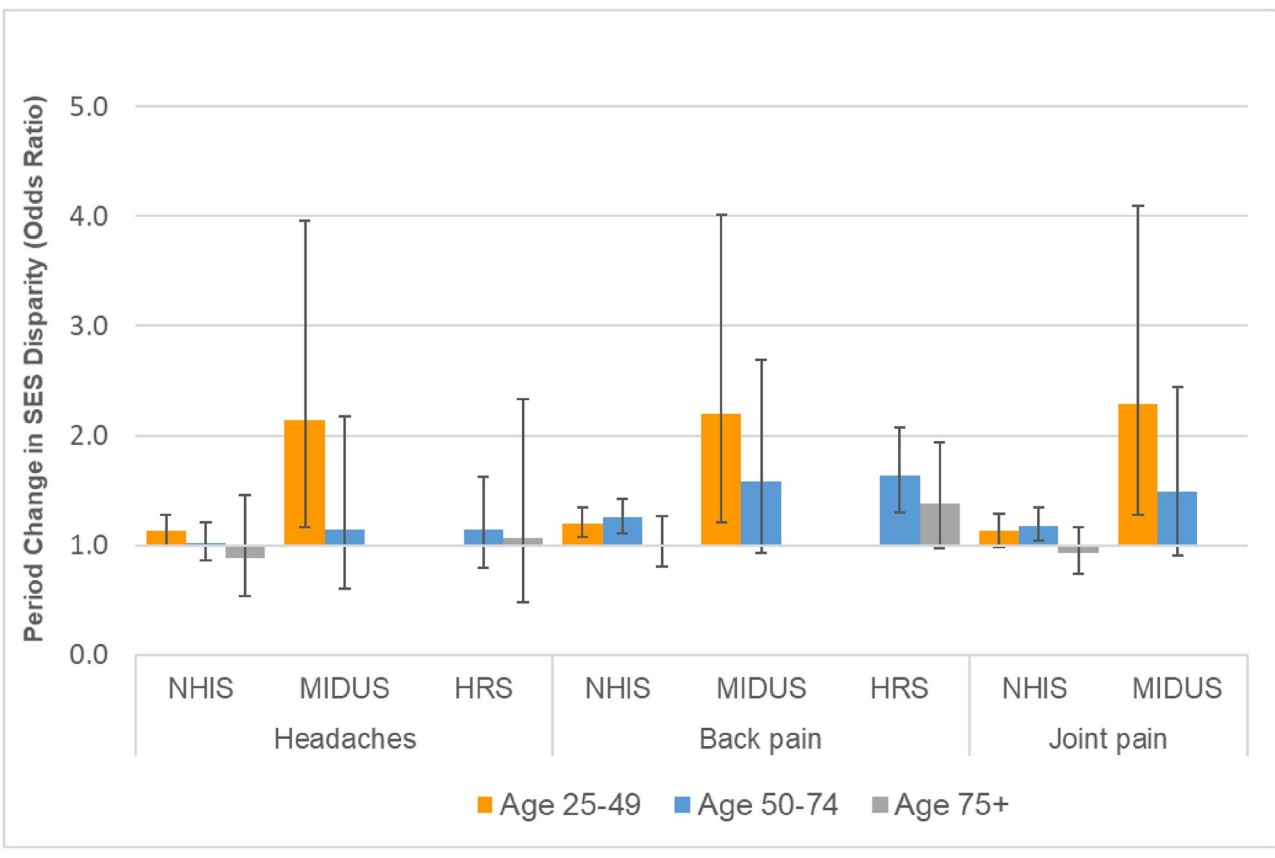

**Fig 8. Changes over time in socioeconomic disparities in specific types of pain by age group.** The SES disparity was computed as the odds ratio for the 90th versus the 10th percentile of SES. For HRS, there were very few respondents aged 75 and older with valid data for headaches or backpain in the 1996 or 1998 waves, when those questions were asked only of new respondents. Thus, the estimates for those outcomes from HRS were based on the odds ratio for 2016–18 relative to 1996–2003. The estimates were based on models fit separately for the three age groups. All models adjusted for sex, age (linear specification), race/ethnicity, period (categorical specification), and SES. The models also included the following 2-way interactions: age x period; age x SES; and period x SES.

[39,40]. Finally, evidence from MIDUS indicated that SES differentials in body mass index and waist circumference widened between 1995–96 and 2011–14 [4]. These inconsistent results cast doubt on the obesity explanation. To explain widening SES disparities in pain and physical limitations, there would have to have been an inverse relationship between SES and the growth in obesity. The only result that is consistent across all surveys and measures of SES is that obesity increased to some degree over time at all levels of SES.

Others have argued that rising pain and declining health may be a consequence of deteriorating social and economic conditions faced by less-educated Americans [1]. However, that hypothesis could not explain the seemingly contradictory results among those with high SES (i.e., among whom overall pain levels increased, but who experienced little change in physical limitations).

Most of the literature presumes that the rise in pain fueled the drug epidemic, but it is possible that the drug epidemic exacerbated the rise in pain. Prolonged use of opioids can increase sensitivity to pain [41]. Even if pain rose, at least in part, because of opioid-induced hyperalgesia, it still would not explain the contradictory results for those with high SES (i.e., if overall pain increased, why was there little change in physical function?). We can only speculate that

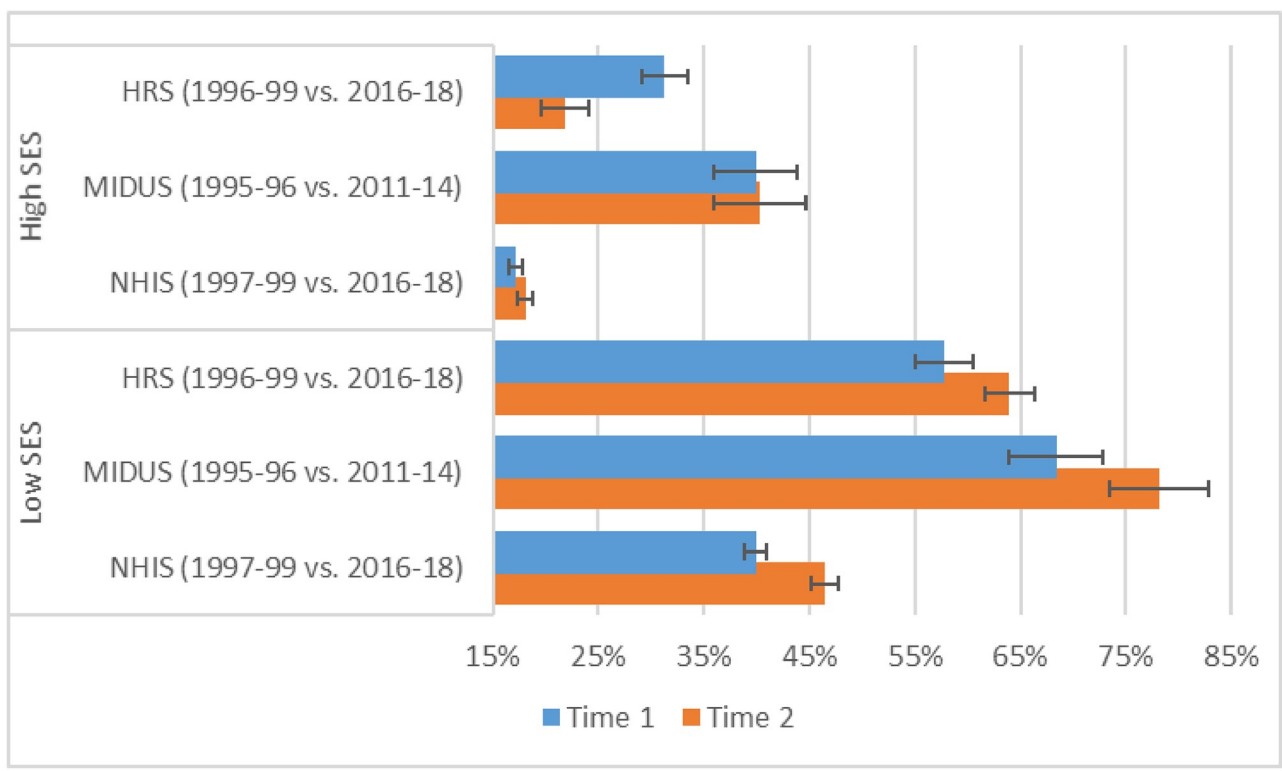

**Fig 9. Predicted prevalence of any physical limitation at age 50 by survey and period for low versus high SES.** These predicted probabilities were based on models that adjusted for sex, age (quadratic specification), race/ethnicity, period (categorical specification), and SES. The models included the following 2-way interactions: age (quadratic) x period; age (quadratic) x SES; and period x SES. They also included a 3-way interaction between age (linear), period, and SES. We computed the probabilities for someone aged 50 at the 10th (low SES) vs. the 90th (high SES) percentiles of SES for the specified periods; all other covariates were fixed at the values observed in the sample.

an increase in pain may have had a greater effect on physical function among disadvantaged Americans because they were more likely to have physically-demanding jobs (e.g., require more walking, lifting, standing for long hours).

## Limitations

Although we have tried to the best of our ability to harmonize the measures, some variation in the measures across surveys remains, which could generate differences in the Results. There were also differences across surveys in the mode of administration: MIDUS used a mail-in, self-administered questionnaire (SAQ), NHIS used in-person interviewing, and HRS used a split mode (about half the interviews were administered in-person and the other half were administered by telephone). Prior work has demonstrated that there is substantial variation across US national surveys in the estimated prevalence of physical limitations at a given age even when the questions seem comparable; those differences may be, at least in part, a result of survey mode [42]. Even if the absolute levels of pain and physical limitation differ by survey because of variation in the measures or survey mode, it would not affect the findings regarding changes in the SES disparities unless the inter-survey differences varied by SES <u>and</u> over time (e.g., perhaps individuals with low SES were more willing to acknowledge pain/limitation in an SAQ than in an in-person interview, whereas reporting by those with high SES was not

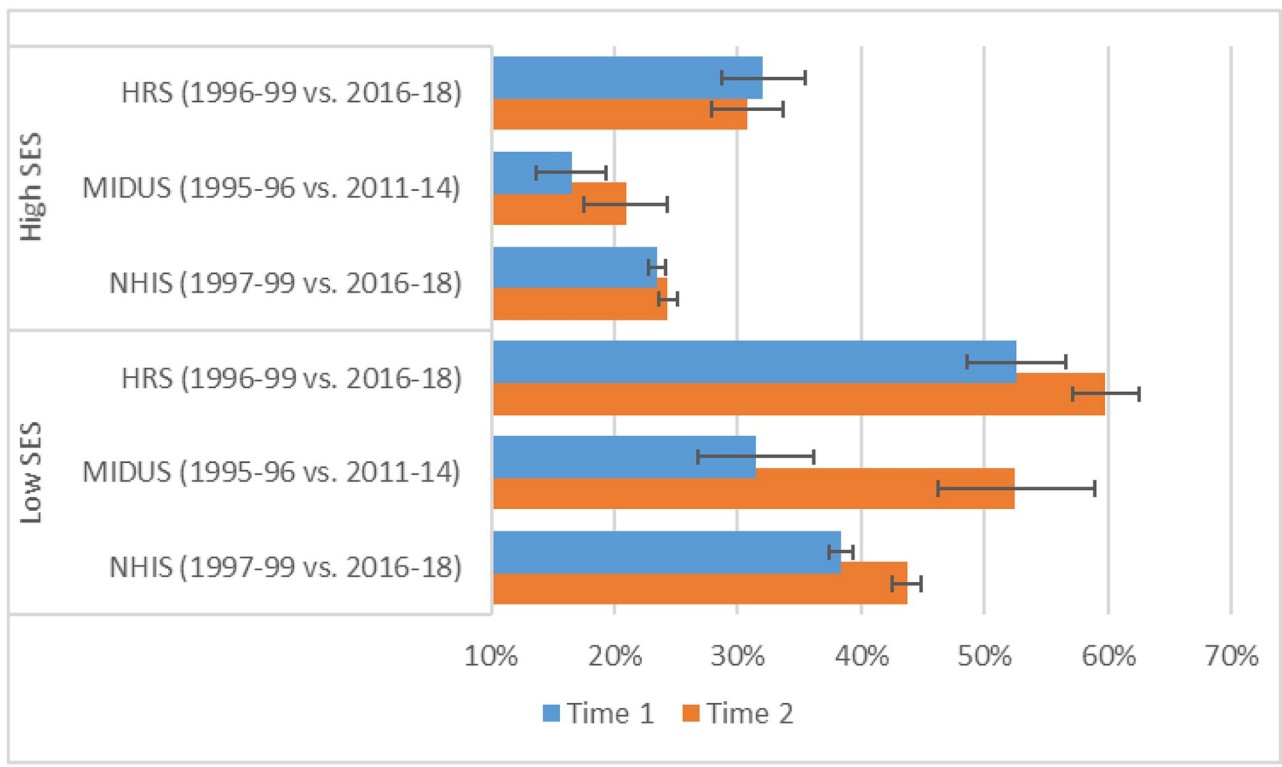

**Fig 10. Predicted prevalence of back pain at age 50 by survey and period for low versus high SES.** These predicted probabilities were based on models that adjusted for sex, age (quadratic specification), race/ethnicity, period (categorical specification), and SES. The models included the following 2-way interactions: age (quadratic) x period; age (quadratic) x SES; and period x SES. They also included a 3-way interaction between age (linear), period, and SES. We compute the probabilities for someone aged 50 at the 10th (low SES) vs. the 90th (high SES) percentiles of SES for the specified periods; all other covariates were fixed at the values observed in the sample.

affected by survey mode). The magnitude of SES widening in physical limitations appears to be largest in HRS and smallest in NHIS with MIDUS in the middle, but it is unclear how variation in survey mode could have produced those differences. A previous study found that respondents reported physical limitations at a much younger age in MIDUS than in NHIS, HRS, or NHANES [42]. If respondents were more likely to report physical limitation in an SAQ, and that differential were especially large for those with low SES and grew larger over time, then we would expect to find that SES widening was greatest in MIDUS. Even when we compare respondents in the same age range across surveys (i.e., 50–74), we still find that the magnitude of SES widening in physical limitations was generally larger in HRS (mix of in-person vs. telephone) than in MIDUS (SAQ only) and smallest in NHIS (in-person only). Unlike physical function, there was little difference between MIDUS and HRS in the estimated SES widening of back pain among those aged 50–74, although the magnitude was slightly bigger in HRS than in NHIS.

Another limitation is that all the measures were based on self-report. Again, even if self-reports were biased, it would not affect the results regarding changes in the SES disparities unless the bias varied by SES <u>and</u> over time. Although some US national surveys (e.g., HRS, NHANES) include performance assessments, the period of collection is not very long, which would hamper our ability to detect period changes in the SES disparity in physical performance.

## Conclusions

Rising pain and growing physical limitations can have important implications not only for quality of life, but also for productivity and the cost of public benefits (e.g., disability, Medicaid/Medicare). It could also exacerbate the drug epidemic. More than 40% of disabled Medicare beneficiaries used an opioid and one-fifth were chronic users of opioids [9]. As Maestas [43, p. 27560] noted, "If less-educated, prime-age Americans are unable to work or are less productive at work because they are in pain, the nation's economic future is at stake. . .to the degree people in pain work less than they would if they were not in pain, they will contribute less in taxes during their working years but need more health care services when they are older." The worst-off Americans are being left behind in a sea of pain and physical infirmity, which will have dire consequences for society as a whole.

## Supporting information

**S1 Appendix. Supplementary material.**
(DOCX)

## Author Contributions

**Conceptualization:** Dana A. Glei, Maxine Weinstein.

**Formal analysis:** Dana A. Glei.

**Funding acquisition:** Maxine Weinstein.

**Methodology:** Dana A. Glei, Maxine Weinstein.

**Project administration:** Maxine Weinstein.

**Software:** Dana A. Glei.

**Visualization:** Dana A. Glei.

**Writing – original draft:** Dana A. Glei.

**Writing – review & editing:** Dana A. Glei, Maxine Weinstein.

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
