## [Decision Letter · Decision Letter 0]

6 Oct 2021

PONE-D-21-24383Disadvantaged Americans are suffering the brunt of rising pain and physical limitationsPLOS ONE

Dear Dr. Glei,

Thank you for submitting your manuscript to PLOS ONE. After careful consideration, we feel that it has merit but does not fully meet PLOS ONE’s publication criteria as it currently stands. Therefore, we invite you to submit a revised version of the manuscript that addresses the points raised during the review process.

Both reviewers found considerable merit with the current manuscript, yet provided feedback on how the manuscript could be improved. Please do your best to address the comments of the reviewers--paying particularly close attention to methodological concerns.

We look forward to receiving your revised manuscript.

Kind regards,

Kenzie Latham-Mintus, PhD, FGSA

Academic Editor

PLOS ONE

2. Please modify the title to ensure that it is meeting PLOS’ guidelines (https://journals.plos.org/plosone/s/submission-guidelines#loc-title). In particular, the title should be "specific, descriptive, concise, and comprehensible to readers outside the field" and in this case we feel it is not informative and specific about your study's scope and methodology

Reviewers' comments:

Reviewer's Responses to Questions

**Comments to the Author**

1. Is the manuscript technically sound, and do the data support the conclusions?

Reviewer #1: Yes

Reviewer #2: Yes

2. Has the statistical analysis been performed appropriately and rigorously? 

Reviewer #1: Yes

Reviewer #2: Yes

3. Have the authors made all data underlying the findings in their manuscript fully available?

Reviewer #1: Yes

Reviewer #2: Yes

4. Is the manuscript presented in an intelligible fashion and written in standard English?

Reviewer #1: Yes

Reviewer #2: Yes

5. Review Comments to the Author

Reviewer #1: Thanks for the opportunity to review the manuscript “Disadvantaged Americans are suffering the brunt of rising pain and physical limitations” (PONE-D-21-24383). The paper utilizes three national datasets to examine the disparities in the development of pain and physical limitation for people in different SES groups. The research design is solid, and the findings are important. I only have several suggestions about the clarification of research goal and method. I raise them below in no particular order.

1) In the introduction part: though I appreciate that the authors stated their hypotheses clearly, it seems a little odd to see hypotheses pop out suddenly without a clear and organized literature review. For example, on the second and third paragraph (line 48 and 56 respectively), hypotheses are mentioned at the beginning of the paragraph before solid evidence from previous literature is discussed.

2) The biggest innovation of the paper is use multidimensional measure of SES, but the innovation and advantages of this measure over other commonly used measures are yet clearly discussed. I suggest use a separate, independent paragraph for this.

3) The aim of using age and period interactions are not clear enough to me. For line 164-165, why including age and period interactions can test “whether rising pain and growing physical limitations are concentrated in midlife”? Also, why only 50-year-old is used to show the period effects given that all three datasets covered 50 and other above 50 aged groups? What if another age is used? This could be added in the sensitivity analysis.

4) Line 169: why HRS data are the only unweighted here?

Reviewer #2: OVERVIEW

In this well-written paper, the authors pursue an important research question carefully and rigorously: Have socioeconomic disparities in pain and physical limitations widened over time in the U.S., and does the answer to this question depend on the age group one is examining? A small but growing number of recent articles have examined temporal trends in pain and pain disparities, but this one directly tackles the interaction between disparities and age group, and carefully examines whether changes in disparities are driven by improving or declining health in each age group.

The authors use 3 different large, national data sets; smart analytic choices; and a number of important robustness checks to answer their research questions with high rigor. For example, by using a relative measure of SES (90th vs. 10th percentile), they circumvent the possibility of lagged selection bias that has plagued some recent work on changes in disparities. The detailed supplementary materials further underscore the authors’ thorough knowledge of their data, variables, and findings. The main weaknesses are relatively minor, and have to do with conceptualization and interpretration; e.g., a lack of clarity about the relationship between pain and functional limitations.

MAJOR COMMENTS

Results: Different age groups often appear to have overlapping confidence intervals in the figures. For example, in Figures 6-8, the CIs for HRS respondents ages 50-74 and 75+ nearly always overlap (although the point estimates are consistently higher for the 50-74 group). Don’t these overlapping CIs weaken the authors’ claim that SES disparities have widened more for the younger group?

Discussion: The latter, speculative half of the Discussion is weaker than earlier (more methodological or descriptive) parts of the text, with several unclear claims. One reason is because the authors are never explicit (in the Discussion or elsewhere) of what they see as the link between pain and functional limitations. They seem to presume a close link between the two--hence they can describe “the contradictory increase in pain but improvement in physical function among those with high SES”--but how close? Functional limitations could reflect problems with fatigue, balance/coordination, cognitive function, etc. as well as problems with pain. Couldn’t pain and function show different patterns because they are reflecting different underlying problems? This should be spelled out. (Additional, more minor comments about the Discussion are below.)

MINOR COMMENTS

Abstract: I suggest specifying the time period under examination early in the abstract.

First page of Introduction (two questions): (1) “least able to bear the consequences” is vague. What kinds of consequences? (2) Authors write, “We hypothesize that the gap in SES disparities has widened more at younger ages (25-49) than among the oldest Americans.” However, elsewhere in the paper, the focus is on “midlife” adults (50-74) vs. older ones (75+), not “younger” adults. E.g., the abstract tells us that disparities “widened even more in midlife than in late life”. For consistency, the authors may wish to clarify that they focus on *3* different age groups, and that they hypothesize and find that disparities are typically larger among younger than older age groups. (Even removing the “(25-49)” from the quoted sentence would help, since then readers aren’t primed to focus specifically on that group vs. all others.)

Discussion: the authors imply that “perceived pain” is somehow different from “real pain”--but *all* pain is “perceived pain”. One could fairly ask whether pain *reporting* styles have changed over time, but to imply that pain exacerbated by opioid-related sensitization is less “real” than other pain is confusing.

Discussion: The quote from Joffe-Walt is intriguing, but also a bit of a non-sequitur in context. I was not clear on its implications in the authors’ view. Is the idea that low-SES individuals are exaggerating/fabricating their degree of functional limitation, in order to quality for disability benefits?

6. PLOS authors have the option to publish the peer review history of their article (what does this mean?). If published, this will include your full peer review and any attached files.

Reviewer #1: No

Reviewer #2: No

---

## [Author Response · Author response to Decision Letter 0]

26 Oct 2021

We have uploaded a separate file that provides a point-by-point response to the reviewers' comments.

---

## [Decision Letter · Decision Letter 1]

1 Dec 2021

Disadvantaged Americans are suffering the brunt of rising pain and physical limitations

PONE-D-21-24383R1

Dear Dr. Glei,

We’re pleased to inform you that your manuscript has been judged scientifically suitable for publication and will be formally accepted for publication once it meets all outstanding technical requirements.

Kind regards,

Kenzie Latham-Mintus, PhD, FGSA

Academic Editor

PLOS ONE

Additional Editor Comments (optional):

Reviewers' comments:

Reviewer's Responses to Questions

**Comments to the Author**

1. If the authors have adequately addressed your comments raised in a previous round of review and you feel that this manuscript is now acceptable for publication, you may indicate that here to bypass the “Comments to the Author” section, enter your conflict of interest statement in the “Confidential to Editor” section, and submit your "Accept" recommendation.

Reviewer #1: All comments have been addressed

Reviewer #2: All comments have been addressed

2. Is the manuscript technically sound, and do the data support the conclusions?

Reviewer #1: Yes

Reviewer #2: Yes

3. Has the statistical analysis been performed appropriately and rigorously? 

Reviewer #1: Yes

Reviewer #2: Yes

4. Have the authors made all data underlying the findings in their manuscript fully available?

Reviewer #1: Yes

Reviewer #2: Yes

5. Is the manuscript presented in an intelligible fashion and written in standard English?

Reviewer #1: Yes

Reviewer #2: Yes

6. Review Comments to the Author

Reviewer #1: (No Response)

Reviewer #2: The authors did an admirable job of taking a strong paper and making it even stronger (and clearer). Their responses show that they know their data, methods, and findings very well. I am satisfied with their modifications to the manuscript.

7. PLOS authors have the option to publish the peer review history of their article (what does this mean?). If published, this will include your full peer review and any attached files.

Reviewer #1: No

Reviewer #2: No

---

## [Editor Report · Acceptance letter]

2 Dec 2021

PONE-D-21-24383R1 

Disadvantaged Americans are suffering the brunt of rising pain and physical limitations 

Dear Dr. Glei:

I'm pleased to inform you that your manuscript has been deemed suitable for publication in PLOS ONE. Congratulations! Your manuscript is now with our production department. 

Kind regards, 

on behalf of

Dr. Kenzie Latham-Mintus 

Academic Editor

PLOS ONE